



# A source mechanism for magnetotail current sheet flapping

Liisa Juusola[1,2], Yann Pfau-Kempf[1], Urs Ganse[1], Markus Battarbee[1], Thiago Brito[1], Maxime Grandin[1], Lucile Turc[1], and Minna Palmroth[1,2]

[1]University of Helsinki, Department of Physics, Helsinki, Finland
[2]Finnish Meteorological Institute, Helsinki, Finland

**Correspondence:** L. Juusola (liisa.juusola@fmi.fi)

**Abstract.** The origin of the flapping motions of the current sheet in the Earth's magnetotail is one of the most interesting questions of magnetospheric dynamics yet to be solved. We have used a polar plane simulation from the global hybrid-Vlasov model Vlasiator to study the characteristics and source of current sheet flapping in the center of the magnetotail. The characteristics of the simulated signatures agree with observations reported in the literature. The flapping is initiated by a hemispherically asymmetric magnetopause perturbation, created by subsolar magnetopause reconnection, that is capable of displacing the tail current sheet from its nominal position. The current sheet displacement propagates downtail at the same pace as the driving magnetopause perturbation. The initial current sheet displacement launches a standing magnetosonic wave within the tail resonance cavity. The travel time of the wave within the local cavity determines the period of the subsequent flapping signatures. Compression of the tail lobes due to added flux affects the cross-sectional width of the resonance cavity as well as the magnetosonic speed within the cavity. These in turn modify the wave travel time and flapping period. The compression of the resonance cavity may also provide additional energy to the standing wave, which may lead to strengthening of the flapping signature. The suggested mechanism could act as a source for kink-like waves that have been observed to be emitted from the center of the tail and to propagate toward the dawn and dusk flanks.

**Keywords.** Magnetosphere and space plasma physics (Magnetotail; Numerical simulation studies; Magnetospheric configuration and dynamics)

## 1 Introduction

The magnetotail current or neutral sheet is a relatively narrow region of lobe magnetic field reversal within a broader plasma sheet region between the tail lobes. A satellite on an orbit passing through this region often observes multiple current sheet crossings, indicating that the sheet moves back and forth across the spacecraft several times. This flapping motion of the current sheet is observed as up to tens of nT variations in the magnetic field Geocentric Solar Magnetospheric (GSM) $B_x$ component, often associated with a change of polarity of $B_x$ (Speiser and Ness, 1967).

The temporal scale of the flapping variations ranges from tens of seconds to tens of minutes (Sergeev et al., 1998). The variations are interpreted as up and down motion of the current sheet with respect to the stationary satellite, such that $\partial B_x/\partial t$ anticorrelates with the GSM north component of the plasma bulk velocity ($V_z$). Sergeev et al. (1998) show example events with $V_z$ varying from some tens of km/s to some hundreds of km/s. The majority of the flapping events are observed during thin





current sheets within $\pm 10\,\mathrm{min}$ around substorm onset or intensification (Sergeev et al., 1998). Laitinen et al. (2007) presented an event during which current sheet oscillations were timed to occur both before tail reconnection and during it.

Two types of flapping motions have been distinguished: steady flapping and kink-like flapping (Rong et al., 2015). Steady flapping refers to oscillations in the $z$ direction during which the current sheet normal remains more or less in the same

direction. Such oscillations do not propagate as waves but remain stationary. The kink-like flapping, on the other hand, is able to propagate as waves. The sheet as a whole is tilted, rather than being locally expanded and contracted like in the sausage wave, which could also have explained the observed flapping signatures (Runov et al., 2003; Sergeev et al., 2003). The kink-like flapping may propagate along the $y$ axis, in which case the current sheet normals for subsequent crossings vary the in $y-z$ plane, or along the $x$ axis, so that the normals vary in $x-z$ plane. All three flapping types (steady flapping, kink-like along $y$,

and kink-like along $x$) can yield similar time series of the $B_x$ component.

Flapping has been observed as close to the Earth as 12 Earth radii ($\mathrm{R_E}$), i.e., close to the transition region or hinge point between tail-like and dipolar field lines (Sergeev et al., 1998). The waves are tail-aligned structures with the cross-tail wavelength ($\sim 5\,\mathrm{R_E}$) clearly smaller than the along-tail length ($> 10\,\mathrm{R_E}$) (Runov et al., 2009), and are observed to propagate toward the flanks from the center of the tail at velocities of some tens of $\mathrm{km/s}$ (Sergeev et al., 2004).

The origin of the flapping motion has not been established, although several tentative explanations, including solar wind variations (e.g., Speiser and Ness, 1967; Sergeev et al., 2008) and internal sources (e.g., Sergeev et al., 2004; Golovchanskaya and Maltsev, 2005; Zelenyi et al., 2009; Davey et al., 2012; Wei et al., 2015) have been suggested. Bursty Bulk Flows (BBFs), produced as outflows from tail reconnection, are among the suggested internal sources (Gabrielse et al., 2008; Erkaev et al., 2009). Statistical results have demonstrated that the occurrence rate of flapping motions is similar to that of BBFs, and peak in

the central part of the magnetotail (Sergeev et al., 2006).

In this study, we analyze a two-dimensional (2D) polar plane simulation produced using the global magnetospheric hybrid-Vlasov model Vlasiator[1]. The simulation is driven by steady solar wind, characterized by high solar wind speed and southward interplanetary magnetic field (IMF). About half an hour after the start of the simulation, tail reconnection begins. The same run has been used earlier to analyze subsolar magnetopause reconnection (Hoilijoki et al., 2017), onset of tail reconnection

(Palmroth et al., 2017), and ion acceleration by flux transfer events in the magnetosheath (Jarvinen et al., 2018). Our aim is to examine the current sheet flapping signatures present in the simulation before and after the onset of tail reconnection, and to determine the driver of the flapping motions. Because the simulation is 2D, we concentrate on the characteristics and source of the waves in the center of the tail. In 3D, they could drive the kink-like waves that are emitted from the center of the tail and propagate dawnward and duskward. The structure of the paper is as follows: the Vlasiator model is described in Section 2 and

the results presented in Section 3. Section 4 contains discussion and Section 5 summarizes the conclusions.

---

[1]http://www.physics.helsinki.fi/vlasiator





## 2  Methods

We use the hybrid-Vlasov model Vlasiator (Palmroth et al., 2013; von Alfthan et al., 2014; Palmroth et al., 2015). In this version
of the simulation, ions are modeled as 5D velocity distribution functions (2D in space, 3D in velocity) that are propagated
in time according to Vlasov's equation. Ampère's law, Faraday's law, and generalized Ohm's law, including the Hall term,
complete the set of equations. Electrons are neglected apart from their charge-neutralizing behavior.

The spatial resolution of the 2D polar plane simulation is $\Delta x = \Delta z = 300$ km or $\sim 0.047$ $R_E$ (1 $R_E$ = 6371 km) and the
velocity space resolution $\Delta v_x = \Delta v_y = \Delta v_z = 30$ km/s. The velocity space in each spatial grid cell covers $v_x = v_y = v_z =$
$\pm 4020$ km/s. The simulation box extends from $x = 300,000$ km or $\sim 47$ $R_E$ on the dayside to $x = -600,000$ km or $\sim -94$
$R_E$ on the nightside. In the north-south direction the box covers $z = \pm 360,000$ km or $\sim \pm 57$ $R_E$. The inner boundary of the
simulation is at the distance of $30,000$ km or $\sim 5$ $R_E$ from the origin. The geomagnetic field is modeled as a 2D line dipole
that is centered at the origin, aligned with the $z$ axis, and scaled toresult in a realistic magnetopause standoff distance (Daldorff
et al., 2014). Thus, the coordinate system is comparable to GSM.

Steady solar wind, characterized by Maxwellian distribution functions, proton density of $1$ cm$^{-3}$, temperature of $0.5$ MK,
velocity of $-750$ km/s along the $x$ axis, and magnetic field of $-5$ nT along the $z$ axis (purely southward IMF), is fed into
the simulation box from its sunward wall ($+x$). Copy-conditions are applied to the other outer walls ($-x$, $\pm z$), and periodic
boundary conditions to the out-of-plane ($\pm y$) directions. The inner boundary enforces a static Maxwellian velocity distribution
and perfect conductor field boundary conditions. The simulation output (moments and fields) is saved every simulated $0.5$ s.

## 3  Results

In this section, we start (section 3.1) by examining the characteristics of the current sheet flapping signatures in the simulation
and by comparing them with the observations cited in the Introduction. After that (section 3.2), we suggest a mechanism that
could explain how the flapping motion is initated and maintained.

### 3.1  Characteristics of current sheet flapping signatures

Figure 1 shows the earthward component of the magnetic field ($B_x$, color) at the nominal location of the magnetotail current
sheet ($y = z = 0$) as a function of $x$ and time (MM:SS). The tilted black lines indicate motion at the solar wind velocity
$V_x = -750$ km/s. The vertical black lines at 19:40 and 27:00 mark the time of Fig. 7 and the onset time of tail reconnection at
$x \approx -14$ $R_E$ (Palmroth et al., 2017), respectively. The most striking feature in the plot is the alternating positive and negative
enhancements of $B_x$ of the order of $10$ nT in amplitude. The signatures first appear in the transition region and subsequently
propagate tailward at a velocity which is generally close to the solar wind velocity as indicated by the tilted black lines. The $B_x$
enhancements can appear fairly regular for a while (e.g., around $x = -40$ $R_E$ between 24:00 and 30:00) or quite irregular (e.g.,
tailward of $x = -50$ $R_E$). The durations of the enhancements at a given location vary from a few minutes to less than a minute,
in agreement with Sergeev et al. (1998). The along-tail lengths of the signatures vary, but can be $> 10$ $R_E$, in agreement with





Runov et al. (2009). The enhancements occur for $\sim$13 min before the onset of tail reconnection as indicated by the vertical black line, and can be observed at least for $\sim$4 min after, in agreement with Sergeev et al. (1998) and Laitinen et al. (2007), before the signatures are disrupted by the spreading effects of tail reconnection. After this, $B_x$ enhancements can still be observed between $-30$ and $-10$ R$_E$ until the end of the simulation, i.e., 9 min after the onset of reconnection.

The $B_x$ signatures resemble those associated with current sheet flapping, discussed in the Introduction. In order to check whether this might indeed be the case, Fig. 2 shows the time derivative of $B_x$ ($\partial B_x/\partial t$) and Fig. 3 the $z$ component of the ion bulk velocity ($V_z$). The cyan curves in Fig. 3 indicate isocontours of pressure in the $z$ direction ($P_{magnetic,z} + P_{thermal,z}$) with a thicker curve indicating higher pressure (levels 0.05 nPa, 0.1 nPa, and 0.2 nPa). The curves have only been plotted in the tail-like region undisturbed by significant effects from tail reconnection. The magenta dots in Fig. 3 mark flapping half periods

at $x = -30$ R$_E$, $x = -40$ R$_E$, and $x = -65$ R$_E$, identified based on sign changes of $V_z$. Comparison of Fig. 2 and Fig. 3 shows that $V_z$ and $\partial B_x/\partial t$ are anticorrelated (this can also be seen in Fig. 6 below), implying that the variations in $B_x$ are caused by up and down motion of the current sheet with respect to the $z = 0$ plane (Sergeev et al., 1998). The amplitude of $V_z$ is also in agreement with the observations of Sergeev et al. (1998).

Fig. 3 illustrates that while the first significant signature of downward ion bulk flow (blue) propagates all the way through the

tail at a speed very close to the solar wind speed, the tailward propagation of the subsequent signatures is disrupted at some $x$ distances (e.g., around $x \approx -55$ R$_E$ at 28:00). These $x$ distances are not constant but they appear to propagate tailward as well, although more slowly than the flapping signatures. Furthermore, the characteristic period of the flapping appears to change at these locations such that signatures closer to the Earth have a smaller characteristic period than those farther down the tail. The characteristic period within a given region also seems to decrease with increasing time. The cyan curves indicating isocontours

of pressure reveal that the changes appear to be related to pressure, such that in regions of higher pressure the period of the signatures is smaller. The pressure increase with increasing time is caused by subsolar reconnection adding semi-open magnetic flux to the lobes before tail reconnection starts to close it efficiently enough (Palmroth et al., 2017).

Fig. 4 and Fig. 5 show the location and thickness of the plasma sheet in the $z$ direction, respectively. The plasma sheet extent in $z$ as a function of $x$ and time was identified as the region between $z = \pm 10$ R$_E$ where the ion thermal pressure

perpendicular to the magnetic field is larger than magnetic pressure ($\beta_\perp = P_{thermal,\perp}/P_{magnetic} > 1$) (e.g., Wang et al., 2006). The $z$ coordinate was obtained as the mean of the largest and smallest plasma sheet $z$ value, and the thickness as their difference. Gray areas in the plots indicate regions where the condition $\beta_\perp > 1$ was not met anywhere between $z = \pm 10$ R$_E$. Comparison of Fig. 4 with Fig. 1 shows good correlation between the plasma sheet motion and $B_x$ enhancements, confirming that the enhancements are indeed produced by up and down motion of the plasma sheet. The extent of the motion in $z$ direction

is not very large, less than 1 R$_E$, in agreement with Speiser and Ness (1967), but because the plasma sheet is thin (Fig. 5), this produces significant changes in $B_x$.

Finally, before moving on to discuss the drivers of plasma sheet flapping, Fig. 6 shows a time series observed by a virtual satellite located at $x = -40$ R$_E$ and $y = z = 0$ in the simulation. From top to bottom, the parameters shown are: $B_x$, $\partial B_x/\partial t$, $V_z$, and plasma sheet $z$ location. The vertical magenta lines identify flapping half periods based on sign changes of $V_z$, and

they correspond to the magenta dots at $x = -40$ R$_E$ in Fig. 3. This plot further clarifies the mutual temporal behavior of the





parameters and may be more straightforward to compare with real satellite observations (e.g., Sergeev et al., 1998) than the color map plots.

## 3.2 A driving mechanism for current sheet flapping

Fig. 7 shows $\beta_\perp$ and magnetic field lines in the $x - z$ plane at 19:40. Closed field lines are drawn in green, semi-open field lines
(with one footpoint at the inner boundary) in black, and open field lines in gray. Field lines that are closed but not attached to the geomagnetic field are drawn in magenta. The tail lobes are estimated to lie between the blue curves. These curves indicate the innermost boundaries where $\beta_\perp > 1$ in the region $|z| > 5$ $\mathrm{R_E}$ and $x < 0$. This proxy is based on the assumption that while the magnetosheath is dominated by the plasma pressure, the tail lobes are magnetically dominated. At the time shown, a hemispherically asymmetric magnetopause perturbation, created around the time when subsolar reconnection first started
to add new semi-open flux tubes to the lobes, has reached the distance $x \approx -40$ $\mathrm{R_E}$ (inside the red box). The asymmetric perturbation consists of a simultaneous compression of the northern tail lobe and expansion of the southern tail lobe. The current sheet between the lobes has been shifted slightly southward from its nominal position at $z = 0$. This shift corresponds to the first strong flapping signature (red, starting around $x \approx -15$ $\mathrm{R_E}$ at $\sim$16:00) in Fig. 1.

Fig. 8 shows $V_z$ as a function of $z$ and time at $y = 0$ and $x = -40$ $\mathrm{R_E}$. The vertical black lines are the same as in Fig. 1.
The white and magenta (only plotted until 31:00) curves indicate the innermost boundaries where $\beta_\perp > 1$ and $\beta_\perp > 0.1$ in the region $|z| > 5$ $\mathrm{R_E}$, respectively. The tail lobes are estimated to lie between the white curves (corresponding to the blue curves in Fig. 7).

Fig. 8 shows that typically $V_z$ is directed toward the plasma sheet in both lobes. Close (roughly within $z = \pm 5$ $\mathrm{R_E}$) to the plasma sheet where the flapping produces alternating positive and negative values of $V_z$ in both hemispheres ($\sim$24:00–30:00).
Close to the magnetopause where localized compressions (e.g., around $z = 23$ $\mathrm{R_E}$ at 14:00–16:00) and expansions (e.g., around $z = 23$ $\mathrm{R_E}$ at 16:00–18:00) cause plasma flow toward and away from the plasma sheet, respectively. Furthermore, after 24:00 the magnetopause starts to expand outward due to the accumulation of magnetic flux recently opened by the subsolar reconnection. In these regions, $V_z$ still reflects the motion of the solar wind and is thus directed away from the plasma sheet. The magenta curves in the plot roughly estimate the boundaries between the pre-existing lobe field lines and the newly added
lobe field lines.

Fig. 8 reveals that the passage of the hemispherically asymmetric magnetopause perturbation at the time when the flapping starts around $x = -40$ $\mathrm{R_E}$ is a unique incident. The simultaneous compression of the northern lobe and expansion of the southern lobe causes a downward plasma flow across the entire tail between $\sim$18:00 and 20:00, which leads to the current sheet shifting downward into the southern hemisphere. Only this first flapping signature appears to have a clear magnetopause
driver. In order to find out what causes the flapping to continue after the initial driver has passed, Fig. 9 shows the time derivative of $B_x$ in the same format as Fig. 8. At the start of the flapping between 18:00 and 20:00, there is a blue signature in both lobes close to the plasma sheet. This is related to positive $B_x$ in the northern lobe weakening and negative $B_x$ in the southern lobe strengthening. Very close to the plasma sheet the signature is positive as the previously close to zero $B_x$ of the current sheet is replaced by positive $B_x$ as the sheet moves downward. For several minutes ($\sim$20:00–24:00) after this clear initial signature the





time derivative of $B_x$ in the lobes consists of small-scale structures, until more coherent, larger-scale signatures associated with strong flapping are established after $\sim 24{:}00$. The small-scale structure of $\partial B_x / \partial t$ in the lobes between the initial magnetopause driver and subsequent establishment of the flapping strongly suggests wave activity between the plasma sheet and the magenta curves.

The magnetotail acts both like a waveguide and a resonance cavity (McPherron, 2005). Alfvén waves that tend to propagate down the tail wave guide along the background lobe magnetic field lines are eventually lost. Magnetosonic waves that propagate perpendicular to the background magnetic field can form standing waves in the tail resonance cavity at a roughly constant distance from the Earth. A tailward propagating displacement of the boundary of the cavity produces a disturbance inside the magnetosphere that stands in the cross-tail direction while propagating down the tail. Fig. 10 shows the magnetosonic speed

($V_{ms}$) across the tail at the distance $x = -40\,\mathrm{R_E}$. There is a distinct change in the magnetosonic speed near the location of the magenta curves. This would explain why the waves if Fig. 9 appear to reflect there. We estimate that the magnetosonic wave resonance cavity lies between these curves.

    Standing waves are only possible at discrete frequencies. The simplest approximation to the fundamental period can be obtained by integrating the magnetosonic travel time $dt = dz/V_{ms}$ across the cavity (or back and forth across one hemisphere

of the cavity). Fig. 11 shows the travel time across half of the magnetotail resonance cavity ($T/2$) at distances $x = -30\,\mathrm{R_E}$ (black), $x = -40\,\mathrm{R_E}$ (red), and $x = -65\,\mathrm{R_E}$ (blue) as a function of time (solid curves). For example, $T/2$ at $x = -40\,\mathrm{R_E}$ has been estimated by integrating $dt = dz/V_{ms}$ over the distance between the magenta curves marked in Fig. 10 and dividing by 2. The dots in Fig. 11 indicate tail flapping half periods with color indicating the corresponding $x$-coordinate. The flapping half periods have been obtained as the time differences between consecutive magenta dots indicated in Fig. 3, and they have been

associated with a time stamp corresponding to the average of the two times. The dashed vertical lines mark the start and end time of the first flapping signature at the three distances.

    The travel time at a given distance decreases with increasing time as the increase in pressure due to the addition of flux tubes opened by subsolar reconnection compresses the lobes, decreasing the cross section of the cavity and increasing the magnetosonic speed within. The bumps in the travel time during the first flapping signature at all three $x$ distances are associated

with the magnetopause perturbation that initiated the flapping. Apart from the first flapping signature at a given distance, there appears to be a good correspondence between the approximated local magnetosonic travel time across half of the resonance cavity and the observed flapping half period. Both Fig. 11 and Fig. 3 show that the period of the flapping signatures increases with increasing distance away from the Earth. At a given location, the period decreases with increasing time. These changes follow the structure and development of pressure indicated in Fig. 3 by the cyan curves. Higher pressure indicates smaller

cross-section of the cavity and higher magnetosonic speed due to the higher magnetic field strength, which is in agreement with the suggestion that the local flapping period is determined by the bounce time of the magnetosonic waves within the cavity.





## 4   Discussion

We have examined current sheet flapping in Vlasiator and suggested a source mechanism that is capable of initiating and maintaining flapping in the central meridian of the magnetotail before and during tail reconnection when the plasma sheet is thin. According to our suggested mechanism, the flapping is initiated by a hemispherically asymmetric perturbation of the magnetopause that is capable of displacing the current sheet from its nominal position. The perturbation travels tailward along the magnetopause (close to the solar wind speed in this case), producing a current sheet displacement that propagates tailward at the same speed. As the first flapping signature is directly driven by the magnetopause perturbation, no correspondence between the duration of flapping signature and the local magnetosonic wave travel time in Fig. 11 is expected. The initial current sheet displacement launches a standing magnetosonic wave within the local resonance cavity. The travel time of the wave within the cavity determines the period of the subsequent flapping signatures, as shown in Fig. 11. Changes in the cross-sectional width of the cavity as well as the magnetosonic speed within the cavity affect the wave travel time and thus the local flapping period. The flapping signatures that can be produced by our suggested mechanism are compliant with the characteristics of plasma sheet flapping in the center of the tail cited in the Introduction.

We do not observe a damping of the current sheet flapping. On the contrary, after the initial displacement, the signature can even strengthen (e.g., at $x = -40$ $R_E$ in 24:00–28:00 in Fig. 1). The reason for this could be the continued compression of the cavity that could provide additional energy to the standing wave.

In our polar plane simulation driven by steady solar wind, the initiating magnetopause perturbation is created by subsolar reconnection. In 3D, such a perturbation could have a finite extent in the $y$ direction, and propagate tailward along the noon-midnight sector magnetopause. Thus, the created plasma sheet flapping in the midnight sector plasma sheet could act as a source for the dawnward and duskward propagating flapping signatures that have been observed by satellites (e.g., Sergeev et al., 2004). A solar wind perturbation, such as a tilted interplanetary shock, might also be able to produce the initial plasma sheet displacement, but probably not in such a localized manner (in $y$), as solar wind structures tend to be of much larger scale sizes. However, the perturbation initiating the flapping need not necessarily be at the magnetopause. Any disturbance capable of displacing the current sheet, a BBF for example, should be able to initiate the flapping.

For the purposes of further validation against satellite observations, our model predicts that, e.g., the flapping period at a given observation location should decrease with increasing pressure in the lobes.

## 5   Conclusions

We have used a polar plane simulation from the global hybrid-Vlasov model Vlasiator, driven by steady southward IMF and fast solar wind, to study the characteristics and source of current sheet flapping signatures in the magnetotail. Because the simulation is 2D, we concentrated on the flapping in the center of the tail. Our main results are as follows:

1. The characteristics of the simulated flapping signatures agree with observations reported in the literature.



2. In the simulation, the flapping is initiated by a hemispherically asymmetric magnetopause perturbation, created by sub-solar reconnection, that is capable of displacing the tail current sheet from its nominal position. The current sheet displacement propagates downtail together with the driving magnetopause perturbation.

3. As the intial current sheet displacement passes, it launches a standing magnetosonic wave within the local tail resonance cavity. The travel time of the wave within the cavity determines the period of the subsequent flapping signatures.

4. Increasing pressure in the tail lobes due to an increasing amount of open magnetic flux added by subsolar reconnection can affect the cross-sectional width of the resonance cavity as well as the magnetosonic speed within the cavity. These in turn affect the wave travel time and flapping period. The compression of the resonance cavity may also provide additional energy to the standing wave, which may lead to strengthening of the flapping signature.

5. The suggested mechanism could act as a source for kink-like waves that are emitted from the center of the tail and propagate toward the dawn and dusk flanks. In 3D, the initial driving magnetopause perturbation created by subsolar reconnection could have a finite dawn-dusk extent, possibly corresponding to the width of the subsolar reconnection line.

*Code availability.* Vlasiator is an open source code released under the GPLv2 license. The code is available at http://github.com/fmihpc/vlasiator.

*Author contributions.* LJ carried out most of the analysis and prepared the manuscript. YP-K participated in running the simulation and development of the analysis methods. UG participated in development of the analysis methods. All co-authors read the manuscript and commented on it.

*Competing interests.* The authors declare that they have no conflict of interest.

*Acknowledgements.* We acknowledge The European Research Council for Starting grant 200141-QuESpace, with which Vlasiator (http://www.physics.helsinki.fi/vlasiator) was developed, and Consolidator grant 682068-PRESTISSIMO awarded for further development of Vlasiator and its use in scientific investigations. We gratefully acknowledge Academy of Finland grants number 267144, 312351, and 309937. PRACE (http://www.prace-ri.eu) is acknowledged for granting us Tier-0 computing time in HLRS Stuttgart, where Vlasiator was run in the HazelHen machine with project number 2014112573. The work of LT is supported by a Marie Sklodowska-Curie Individual Fellowship (#704681).





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



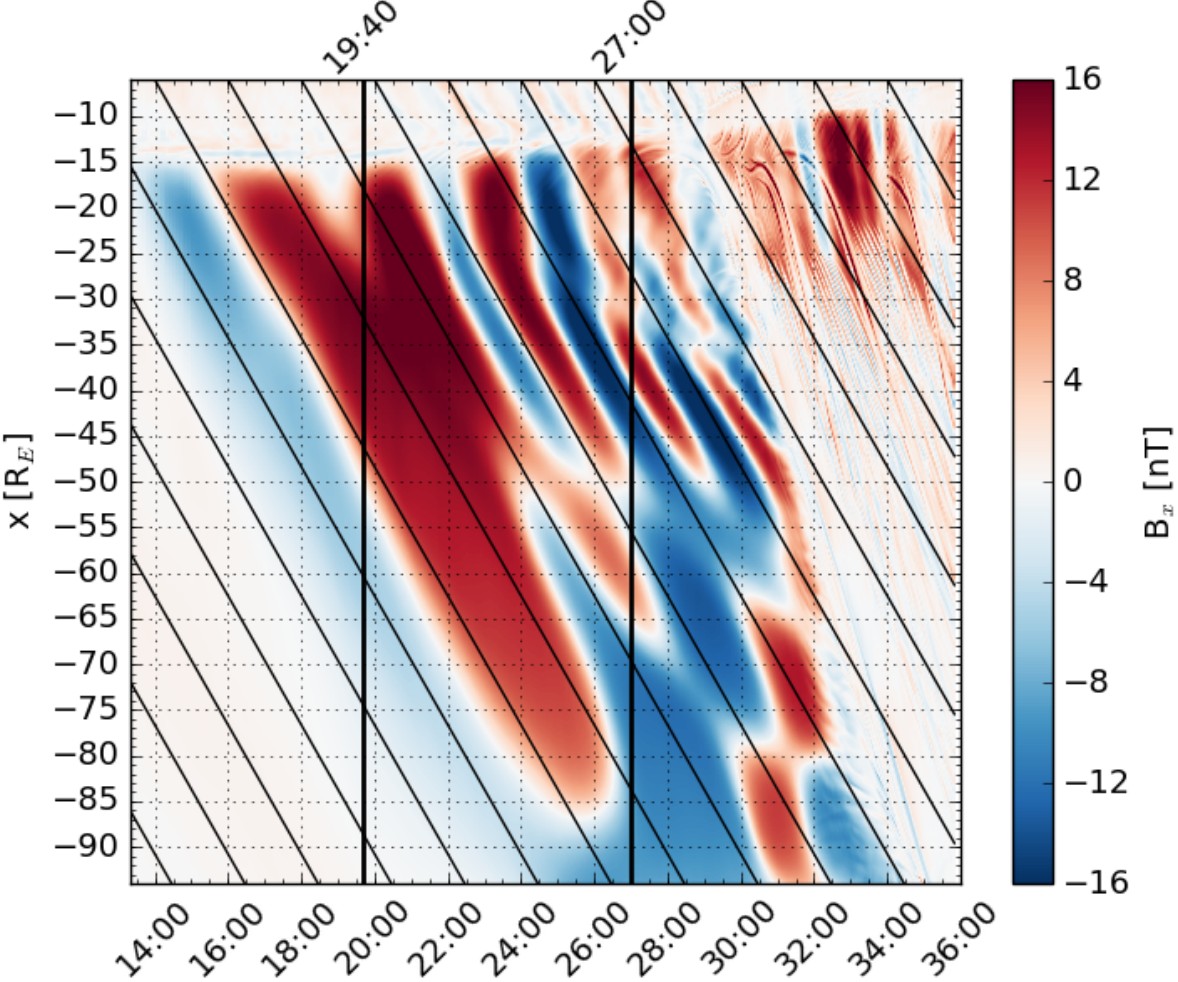

**Figure 1.** Earthward component of the magnetic field ($B_x$, color) at the nominal position of the tail current sheet ($y = z = 0$) as a function of $x$ and time (MM:SS). The tilted black lines indicate motion at the solar wind velocity $V_x = -750$ km/s. The vertical black lines at 19:40 and 27:00 mark the time of Fig. 7 and the onset time of tail reconnection at $x \approx -14$ $R_E$, respectively.



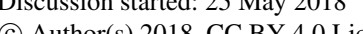



**Figure 2.** The same as Fig. 1 except that the color shows the time derivative of $B_x$. The color scale has been saturated to better show the relevant structures.




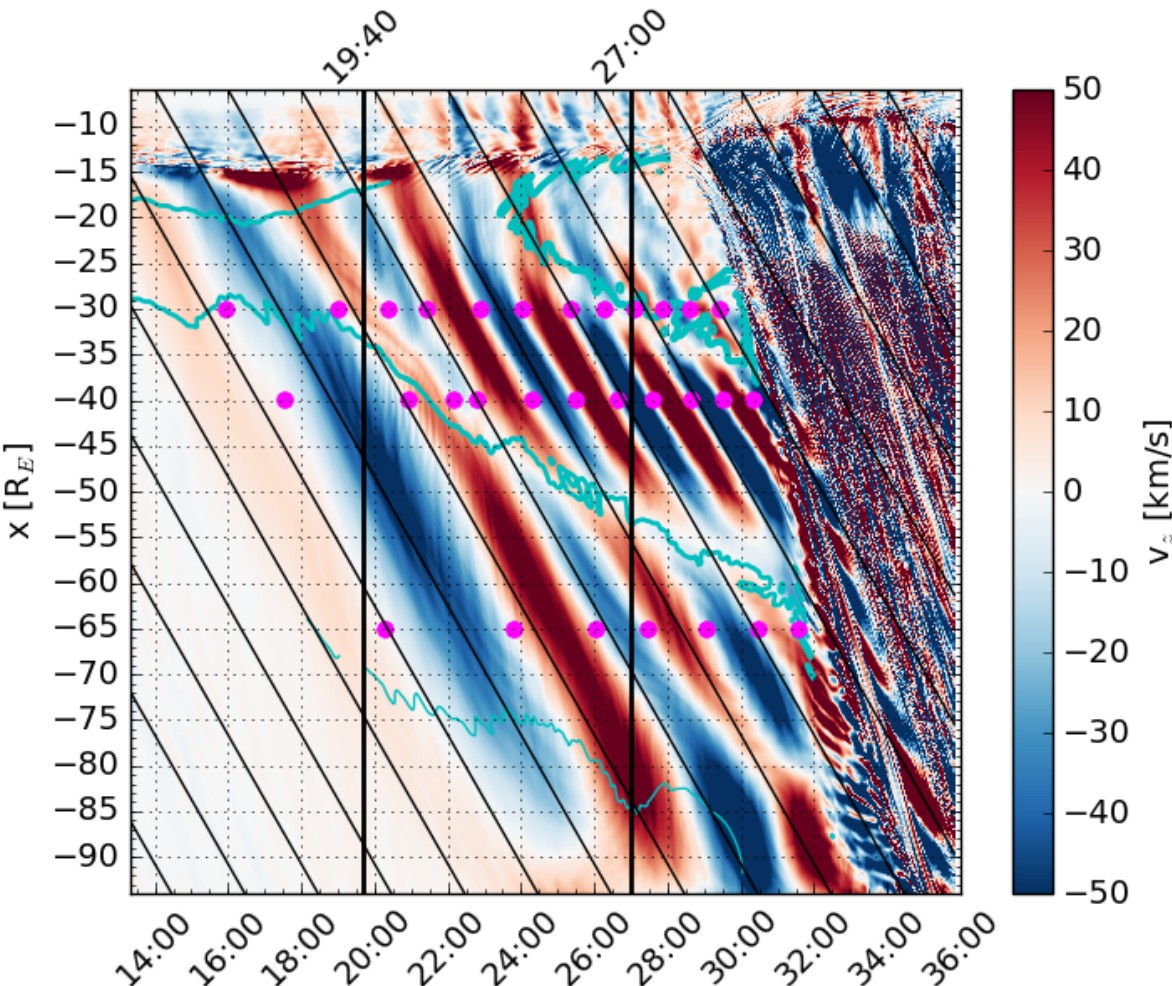

**Figure 3.** The same as Fig. 1 except that the color shows the north component of the ion bulk velocity ($V_z$). The cyan curves indicate isocontours of pressure in the $z$ direction ($P_{magnetic,z} + P_{thermal,z}$) with a thicker curve indicating higher pressure: 0.05 nPa, 0.1 nPa, and 0.2 nPa. The magenta dots mark identified flapping half periods at $x = -30\,\mathrm{R_E}$, $x = -40\,\mathrm{R_E}$, and $x = -65\,\mathrm{R_E}$.




**Figure 4.** The same as Fig. 1 except that the color shows the location of the plasma sheet center in the $z$ direction. Gray areas indicate regions where the location could not be determined.







**Figure 5.** The same as Fig. 4 except that the color shows the plasma sheet thickness.



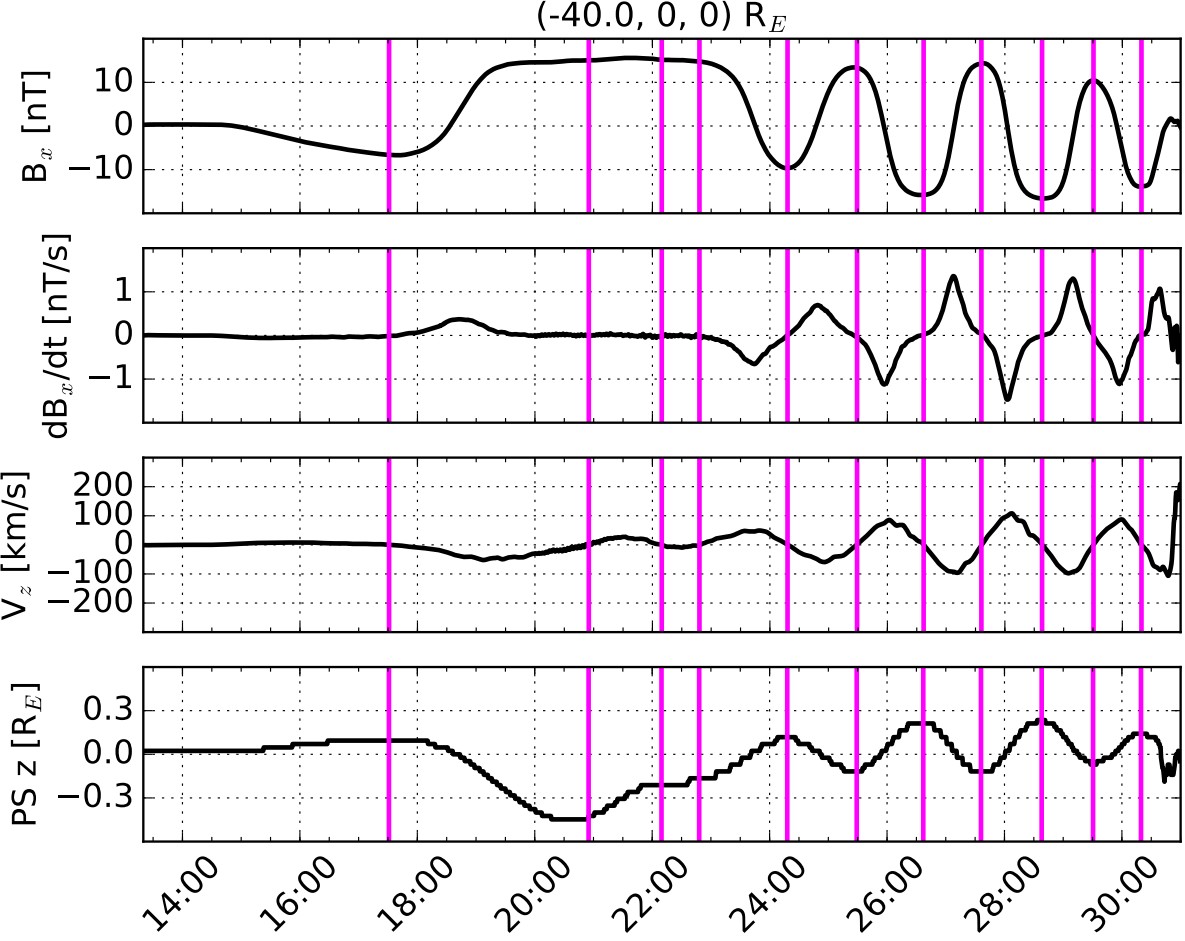

**Figure 6.** Time series observed by a virtual satellite located at $x = -40$ $R_E$ and $y = z = 0$ in the simulation. From top to bottom, the parameters shown are: Earthward component of the magnetic field ($B_x$, cf. Fig. 1), time derivative of $B_x$ ($\partial B_x / \partial t$, cf. Fig. 2), north component of the ion bulk flow ($V_z$, cf. Fig. 3), and location of the plasma sheet center in the $z$ direction (PS $z$, cf. Fig. 4). The vertical magenta lines identify flapping half periods based on sign changes of $V_z$, and they correspond to the magenta dots at $x = -40$ $R_E$ in Fig. 3.





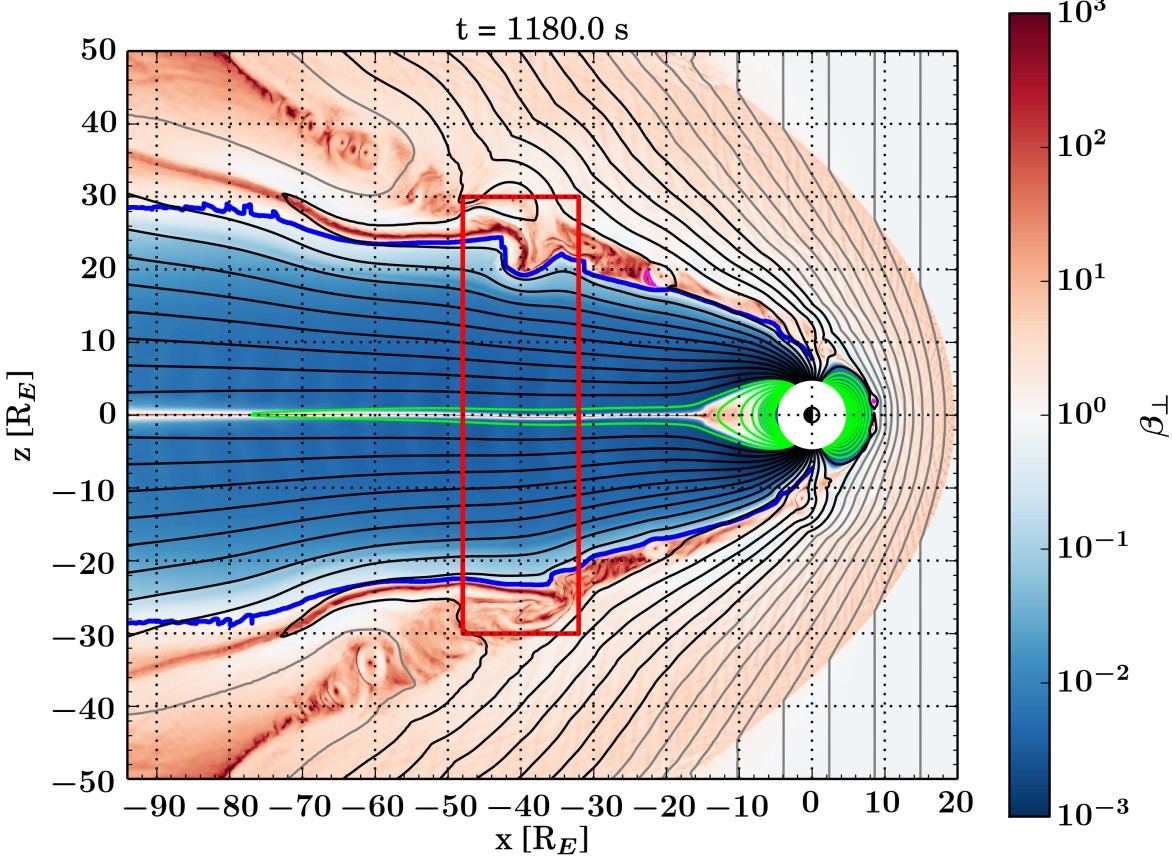

**Figure 7.** Ratio of ion thermal pressure prependicular to the magnetic field and magnetic pressure $\beta_\perp = P_{thermal,\perp}/P_{magnetic}$ and magnetic field lines in the $x - z$ plane at 19:40 (1180.0 s). Closed field lines are drawn in green, semi-open field lines in black, and open field lines in gray. Field lines that are closed but not attached to the geomagnetic field are drawn in magenta. The tail lobes are estimated to lie between the blue curves. The curves indicate the innermost boundaries where $\beta_\perp > 1$ in the region $|z| > 5$ $R_E$ and $x < 0$. Inside the red box, a hemispherically asymmetric magnetopause perturbation compresses the northern tail lobe and expands the southern tail lobe, causing the current sheet between the tail lobes around $x = -40$ $R_E$ to shift slightly southward from its nominal position at $z = 0$.





**Figure 8.** North component of the ion bulk velocity ($V_z$) as a function of $z$ and time at $y = 0$ and $x = -40$ $\mathrm{R_E}$. The vertical black lines are the same as in Fig. 1. The white and magenta (only plotted until 31:00) curves indicate the innermost boundaries where $\beta_\perp > 1$ and $\beta_\perp > 0.1$ in the region $|z| > 5$ $\mathrm{R_E}$, respectively. The tail lobes are estimated to lie between the white curves and the magnetosonic wave resonance cavity between the magenta curves.



**Figure 9.** The same Fig. 8 except that the color shows the time derivative of $B_x$.





**Figure 10.** The same Fig. 8 except that the color shows the magnetosonic speed ($V_{ms}$).



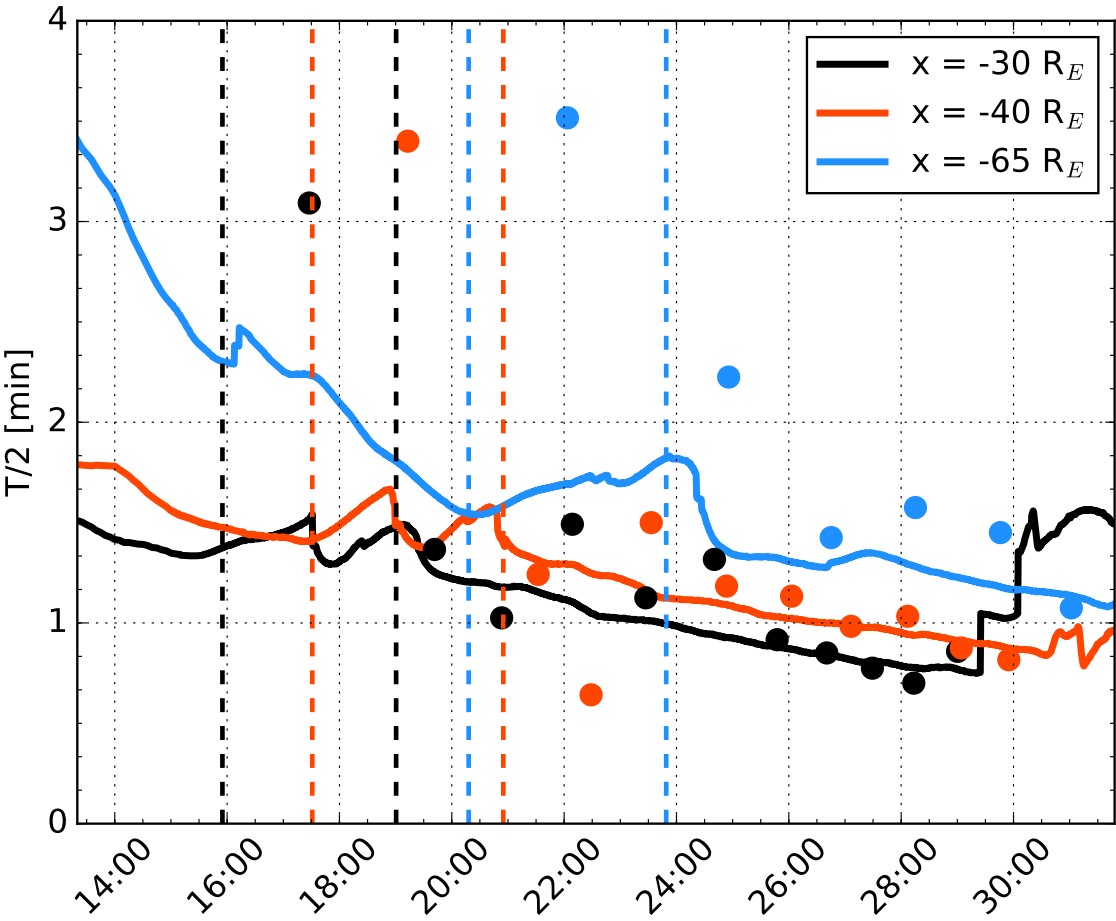

**Figure 11.** Travel time of magnetosonic waves across half of the magnetotail resonance cavity ($T/2$) at distances $x = -30$ R$_E$ (black), $x = -40$ R$_E$ (red), and $x = -65$ R$_E$ (blue) as a function of time (solid curves). For example, $T/2$ at $x = -40$ R$_E$ has been estimated by integrating $dt = dz/V_{ms}$ over the distance between the magenta curves marked in Fig. 10 and dividing by 2. The dots indicate tail flapping half periods with color indicating the corresponding $x$-coordinate. The flapping half periods have been obtained as the time differences between consecutive magenta dots indicated in Fig. 3, and they have been associated with a time stamp corresponding to the average of the two times. The dashed vertical lines show the start and end time of the first flapping signature at the three distances.