# Peer review of "A source mechanism for magnetotail current sheet flapping"

_Annales Geophysicae, 2018_

## Referee Comment (RC1) · Anonymous Referee #1 · 18 Jun 2018

Using the two dimensional global hybrid-Vlasov model Vlasiator, Juusola et al studied the characteristics and source of current sheet flapping in the center of the magnetotail. Their simulations show that an initial down-tail propagating current sheet displacement caused by a hemispherically asymmetric magnetopause perturbation can launch a standing magnetosonic wave within the magnetotail, which acts as a resonance cavity, creating subsequent flapping waves in the current sheet. In three dimensional, Juusola et al suggest that such source mechanism for current sheet flapping could create kink-like waves that had been observed to emit from the center of the tail towards the dawn-dusk flanks. The simulation results from this study could potentially provide explanation on the mechanism for current sheet flapping, which till this day remains unknown, and increase our understanding of the tail flapping phenomenon. However,

much clarification is needed as terminologies are not clearly defined and loosely used. Analysis of results to support their conclusion are lacking, vague and qualitative. In my opinion, major revisions to the manuscript and further clarifications are required.

Comments: 1. Page 1 Line 19: Shouldn't it be "up and down relative to the spacecraft", instead of "back and forth across the spacecraft"? Please clarify.

2. Page 2 Line 9: Insert appropriate reference Sun et al., [2013] THEMIS Observation of a magnetotail current sheet flapping.

3. Page 2 Line 9: In this sentence, the authors categorized current sheet flapping events into three types (Steady, kink-like along y and kink-like along x). To my understanding of this study, Juusola et al focused primarily on the "kink-type along x" current sheet flapping. However, subsequently in the text, the authors used the word current sheet flapping to describe tailward propagating displacement of the current sheet, which I take it to mean "kink-type along x". However, the term current sheet flapping is more commonly described as steady or kink-like along the y-direction in current literature. The authors should clearly state or define what kind of current sheet flapping they are referring to throughout the text so as to not confuse the readers. If at all possible, I would suggest the authors to avoid using tail flapping in this study since many current sheet flapping studies using THEMIS and Cluster data concluded that the current sheet flapping waves travel towards the dawn or dusk flanks (e.g. See review paper by M. Volwerk), which to my understanding from the text, is not what the authors are referring to. This will avoid confusing the readers.

4. Page 3 Line 13: The authors should justify their choice of solar wind parameters in their simulation. A particular set of solar wind conditions, instead of a range of values, are used in this study. This begs the question of how does the solar wind conditions affect the simulation results and conclusion of this study.

5. Page 3 Line 23: I strongly suggest that the authors start section 3 with the simulation results shown in Figure 7. By replacing Figure 1 with Figure 7, it will provide context for

readers who are either not used to or not familiar with simulation studies and improve
the flow of the manuscript.

6. Page 5 Line 10: One of the main conclusions of this simulation study was that the
"asymmetric perturbation consists of a simultaneous compression of the northern tail
lobe and expansion of the southern tail lobe" drives current sheet flapping as shown in
their simulation results. However, it is unclear whether this asymmetric perturbation in
the simulation is physical or numerical. Furthermore, the authors mentioned that this
asymmetric perturbation is caused by subsolar magnetic reconnection (line 8), which
is counter-intuitive. Under steady solar wind conditions and dayside reconnection oc-
curring at the subsolar magnetopause region, shouldn't the loading of the open flux in
the two hemisphere of the tail be equal? One might think that unequal loading of open
flux in the northern and southern tail lobe is caused by dayside reconnection occur-
ring at higher latitude. Would this implies that the perturbation is a numerical effect?
Furthermore, Figure 3 shows that there are regions of high beta around the nightside
magnetopause surface. Are there turbulence occurring on the magnetopause surface?
Could that been the cause of the asymmetric perturbation? Please clarify.

7. Page 6 Line 9: The use of "cross-tail direction", which traditionally referred to the y-
direction, is very confusing. The simulation is two dimensional in the x and z-direction.
Unless the authors meant cross-tail in the z-direction? If that's the case, the authors
should be clear on that as ambiguous use of words could mislead the readers.

8. Page 7 Line 1: In the discussion section, Juusola et al suggested that in three di-
mensions, the asymmetric perturbation could have a finite extent in the y-direction, thus
driving current sheet flapping in the dawn-dusk direction. However, the authors neither
substantiate their conclusion with any 3D simulations results nor conduct any experi-
ments that investigate the effects of a finite IMF By on the occurrence and properties
of tail flapping waves. Since the authors listed this as one of the five main conclusion
of this study, I think substantial work should be done to demonstrate the connection
between asymmetric perturbation mechanism and current sheet flapping in the dawndusk direction as observed by earlier studies, rather than simply providing a hand-wavy, qualitative explanation.

9. Page 7 Line 25: Juusola et al stated model predictions for the purpose of future validations with satellite observations. However, the authors did not follow up on this idea of validations with observations, which I think is a wonderful idea. If it is the authors' intention to validate their simulation results with observations, this study should provide more quantitative results (e.g. what is the relationship between flapping period and lobe pressure? Does it follows a power law or linear relation etc?) and measurable quantities. These information could be easily obtained from the simulation.

---

## Referee Comment (RC2) · Anonymous Referee #2 · 23 Jun 2018

Using the 2-D global hybrid-Vlasov model Vlasiator, authors studied the response of magnetotail at the tail center to the magnetopause perturbation, created by subsolar magnetopause reconnection. Authors declared that the appeared oscillation of tail Bx component should be the kink-like flapping motion propagating towards both tail flanks. Nonetheless, the simulation is 2-D, The variation of this osscillation in Y direction can't be investigated, it is still hard to convince readers that the oscillation of Bx component is indeed associated with the kink-like flapping propagating azimuthally. I will explain my reasons in the following.

Major comment As stated in the introduction, Rong et al.(2015) found that the different flapping modes can yield a same flapping sequence of Bx component. How can you differentiate these flapping modes in your simulation? In my view, only the 3-D

simulation can unambiguously answer it.

Specific comments 1. Line 16 of page 2, the paper of Shen et al.[2008,AG] and Forsyth et al.,(2009,AG) should be cited when referring the disturbance of solar wind flow as the flapping source. Shen, C., Z. J. Rong, X. Li, M. Dunlop, Z. X. Liu, H. V. Malova, E. Lucek, and C. Carr (2008), Magnetic configurations of tail tilted current sheet, Ann. Geophys., 26, 3525–3543. Forsyth, C., M. Lester, R. C. Fear, E. Lucek, I. Dandouras, A. N. Fazakerley, H. Singer, and T. K. Yeoman (2009), Solar wind and substorm excitation of the wavy current sheet, Ann. Geophys., 27, 2457–2474.

2. The joint observation of flapping event by TC-1 and Cluster (Zhang et al., 2005,AG) showed that the kink-like flapping waves propagate longitudinally with the same flapping phase at different X coordinates. However, it can not be characterized in your 2-D simulation, e.g. Fig1. Zhang, T. L., et al. (2005), Double Star/Cluster observation of neutral sheet oscillations on 5 August 2004, Ann. Geophys., 23, 2909–2914.

3. Fig.3 predicts a dispersive flapping waves with time-increased frequency. To my knowledge, there is no observation evidence to back up it. Careful comparisons are needed.

4. Even your 2-D simulation is valid to explain the triggering of kink-like flapping, the magnetopause disturbance is not the unique source. What I mean is that, the sources result in the pressure imbalance over tail current sheet could be multiple.

---

## Referee Comment (RC3) · Anonymous Referee #1 · 26 Jun 2018

Dear Juusola et al.,

Thank you for your replies to my comments on the manuscript. It adequately answers my questions and concerns. I have no further questions or concerns. It is an interesting study on current sheet flapping and I look forward to reading future studies on this research topic.

Sincerely
* * *

---

## Author Comment (AC1) · 26 Jun 2018

>Anonymous Referee #1

>

>Using the two dimensional global hybrid-Vlasov model Vlasiator, Juusola et al studied the characteristics and source of current sheet flapping in the center of the magnetotail. Their simulations show that an initial down-tail propagating current sheet displacement caused by a hemispherically asymmetric magnetopause perturbation can launch a standing magnetosonic wave within the magnetotail, which acts as a resonance cavity, creating subsequent flapping waves in the current sheet. In three dimensional, Juusola et al suggest that such source mechanism for current sheet flapping could create

kink-like waves that had been observed to emit from the center of the tail towards the dawn-dusk flanks. The simulation results from this study could potentially provide explanation on the mechanism for current sheet flapping, which till this day remains unknown, and increase our understanding of the tail flapping phenomenon. However, much clarification is needed as terminologies are not clearly defined and loosely used. Analysis of results to support their conclusion are lacking, vague and qualitative. In my opinion, major revisions to the manuscript and further clarifications are required.

We thank the Referee for their constructive comments. We are happy to make many of the suggested changes. In some cases we suggest alternative approaches, in order to avoid making the manuscript too long to be concise and readable. Please see below for our point-by-point replies.

>Comments: 1. Page 1 Line 19: Shouldn't it be "up and down relative to the space-craft", instead of "back and forth across the spacecraft"? Please clarify.

Yes, we are happy to make the suggested correction.

>2. Page 2 Line 9: Insert appropriate reference Sun et al., [2013] THEMIS Observation of a magnetotail current sheet flapping.

We can add the reference.

>3. Page 2 Line 9: In this sentence, the authors categorized current sheet flapping events into three types (Steady, kink-like along y and kink-like along x). To my under-standing of this study, Juusola et al focused primarily on the "kink-type along x" current sheet flapping. However, subsequently in the text, the authors used the word cur-rent sheet flapping to describe tailward propagating displacement of the current sheet, which I take it to mean "kink-type along x". However, the term current sheet flapping is more commonly described as steady or kink-like along the y-direction in current litera-ture. The authors should clearly state or define what kind of current sheet flapping they are referring to throughout the text so as to not confuse the readers. If at all possible,

[Figure]

I would suggest the authors to avoid using tail flapping in this study since many current sheet flapping studies using THEMIS and Cluster data concluded that the current sheet flapping waves travel towards the dawn or dusk flanks (e.g. See review paper by M. Volwerk), which to my understanding from the text, is not what the authors are referring to. This will avoid confusing the readers.

As explained in the first paragraph of the Introduction (page 1, lines 17-21), the term current sheet flapping has originally been used to refer to up and down motion of the current sheet that can be observed as variations in Bx. We show that our simulated signatures produce the appropriate Bx time series, and thus using the term "current sheet flapping" should be justified.

In later studies, different types of flapping (all of which can produce similar time series of Bx) have been distinguished (page 2, lines 3-10). These include the kink-like flapping in the y direction. Because this kind of flapping has been so widely studied, as also pointed out by the Referee, we wanted to make it clear to the reader that our analysis does not directly apply to it, although the results could be relevant to it as well (page 2, lines 27-29).

In order to clarify the issue, we suggest to modify the text on page 2, lines 27-29 to: "Because the simulation is 2D, we concentrate on the characteristics and source of the waves in the center of the tail (i.e., waves in the x-z plane). We also discuss the possibility that in 3D, they could drive the kink-like waves that are emitted from the center of the tail and propagate dawnward and duskward (i.e., waves in the y-z plane)."

>4. Page 3 Line 13: The authors should justify their choice of solar wind parameters in their simulation. A particular set of solar wind conditions, instead of a range of values, are used in this study. This begs the question of how does the solar wind conditions affect the simulation results and conclusion of this study.

A particular set of solar wind conditions instead of a range of values was used because of the heavy computational load of running this type of a simulation. This same run has

been analyzed in several previous studies as well (page 2, lines 23-25). The run was suitable for our study, because current sheet flapping occurred in it. How solar wind conditions affect the results is a very interesting question indeed. However, it is outside the scope of this study.

>5. Page 3 Line 23: I strongly suggest that the authors start section 3 with the simulation results shown in Figure 7. By replacing Figure 1 with Figure 7, it will provide context for readers who are either not used to or not familiar with simulation studies and improve the flow of the manuscript.

We are happy to make the suggested change.

>6. Page 5 Line 10: One of the main conclusions of this simulation study was that the "asymmetric perturbation consists of a simultaneous compression of the northern tail lobe and expansion of the southern tail lobe" drives current sheet flapping as shown in their simulation results. However, it is unclear whether this asymmetric perturbation in the simulation is physical or numerical. Furthermore, the authors mentioned that this asymmetric perturbation is caused by subsolar magnetic reconnection (line 8), which is counter-intuitive. Under steady solar wind conditions and dayside reconnection occurring at the subsolar magnetopause region, shouldn't the loading of the open flux in the two hemisphere of the tail be equal? One might think that unequal loading of open flux in the northern and southern tail lobe is caused by dayside reconnection occurring at higher latitude. Would this implies that the perturbation is a numerical effect? Furthermore, Figure 3 shows that there are regions of high beta around the nightside magnetopause surface. Are there turbulence occurring on the magnetopause surface? Could that been the cause of the asymmetric perturbation? Please clarify.

Loading of open flux in the two hemispheres should indeed be equal under steady southward solar wind conditions. However, the loading process can still create hemispherically asymmetric perturbations. As shown by Hoilijoki et al. (2017) and Jarvinen et al., (2018), the perturbations created by subsolar reconnection (magnetic islands,

exhaust jets, waves) in the simulation are hemispherically asymmetrical at any given moment of time. The instabilities are seeded by noise at the numerical level, which is not symmetrical. Furthermore, as pointed out by the Referee, turbulence can create and strengthen the asymmetric perturbation as well. Thus, the hemispherically asymmetric magnetopause perturbation that we interpret to initiate the flapping can, according to our understanding, be interpreted to be of physical origin. Because we do not believe that the exact creation mechanism of the perturbation is relevant to the results (page 7, lines 17-24), we have omitted any further analysis of its creation from the text.

>7. Page 6 Line 9: The use of "cross-tail direction", which traditionally referred to the y-direction, is very confusing. The simulation is two dimensional in the x and z-direction. Unless the authors meant cross-tail in the z-direction? If that's the case, the authors should be clear on that as ambiguous use of words could mislead the readers.

We are happy to replace "cross-tail" with "x-z".

>8. Page 7 Line 1: In the discussion section, Juusola et al suggested that in three dimensions, the asymmetric perturbation could have a finite extent in the y-direction, thus driving current sheet flapping in the dawn-dusk direction. However, the authors neither substantiate their conclusion with any 3D simulations results nor conduct any experiments that investigate the effects of a finite IMF By on the occurrence and properties of tail flapping waves. Since the authors listed this as one of the five main conclusion of this study, I think substantial work should be done to demonstrate the connection between asymmetric perturbation mechanism and current sheet flapping in the dawn-dusk direction as observed by earlier studies, rather than simply providing a hand-wavy, qualitative explanation.

We agree with the Referee that listing such claims as conclusions would require further analysis, and we suggest to remove the last point from the list of conclusions in section 5. However, we find these to be valid discussion points, the examination of which

could lead to further studies. Thus, we would still like to mention the possibility that the flapping created through our suggested mechanism could act as a source for the waves that propagate in the y direction both in the abstract and in section 5 (as a separate paragraph below the list of conclusions). However, we would further emphasize that this is only a suggestion and would require further study. For example: "The suggested mechanism could act as a source for kink-like waves that are emitted from the center of the tail and propagate toward the dawn and dusk flanks. However, further research using a 3D simulation will be needed to examine this suggestion." Page 1, lines 12-13 could be modified to: "It may be possible that the suggested mechanism could act as a source for kink-like waves that have been observed to be emitted from the center of the tail and to propagate toward the dawn and dusk flanks."

>9. Page 7 Line 25: Juusola et al stated model predictions for the purpose of future validations with satellite observations. However, the authors did not follow up on this idea of validations with observations, which I think is a wonderful idea. If it is the authors' intention to validate their simulation results with observations, this study should provide more quantitative results (e.g. what is the relationship between flapping period and lobe pressure? Does it follows a power law or linear relation etc?) and measurable quantities. These information could be easily obtained from the simulation.

We agree that it may be desirable to derive more quantitative predictions for the purpose of further validations with observations. However, providing numbers for the validation is not straightforward, because they are likely to depend not only on the driving solar wind conditions but the history of the magnetospheric dynamics. A simple confirmation that the period of the flapping signatures decreases as the lobe pressure increases would be a good starting point, and numbers could be provided when running several 3D simulations representing a range of solar wind conditions becomes possible, which would probably require a full dedicated study.

---

## Author Comment (AC2) · 26 Jun 2018

>Anonymous Referee #2

>

>Using the 2-D global hybrid-Vlasov model Vlasiator, authors studied the response of magnetotail at the tail center to the magnetopause perturbation, created by subsolar magnetopause reconnection. Authors declared that the appeared oscillation of tail Bx component should be the kink-like flapping motion propagating towards both tail flanks. Nonetheless, the simulation is 2-D, The variation of this osscilation in Y direction can't be investigated, it is still hard to convince readers that the oscillation of Bx component is indeed associated with the kink-like flapping propagating azimuthally. I will explain
my reasons in the following.

We thank the Referee for their comments and drawing our attention to the fact that some clarification of the text is needed: our intent was not to declare that the simulated oscillations are the kink-type wave that propagates in the dawn-dusk direction but waves in the x-z plane. It was an item of discussion that these waves could maybe act as a source for the waves that propagate in the y direction. Please see below for our detailed replies to the comments and suggested modifications to the text.

>Major comment >As stated in the introduction, Rong et al.(2015) found that the different flapping modes can yield a same flapping sequence of Bx component. How can you differentiate these flapping modes in your simulation? In my view, only the 3-D simulation can unambiguously answer it.

As the simulation is 2D in the x-z plane it cannot contain the kind of waves that propagate in the y direction. Rong et al. (2015) mention two kinds of waves in the x-z plane. It should be possible to analyze the normal directions of the waves to separate these, but we do not consider this to be relevant to our conclusions. We analyze the wave signatures in the x-z plane, determine a mechanism that can start and maintain these waves, and then discuss the possibility that in 3D our suggested mechanism could function in the midnight sector. We suggest that, in 3D, these waves could act as a source for the waves that are emitted from the midnight sector and propagate in the dawn-dusk direction. In order to make this more clear, we would suggest to modify page 2, lines 27-29 to: "Because the simulation is 2D, we concentrate on the characteristics and source of the waves in the center of the tail (i.e., waves in the x-z plane). We also discuss to possibility that in 3D, they could drive the kink-like waves that are emitted from the center of the tail and propagate dawnward and duskward (i.e., waves in the y-z plane)." We have also suggested some further clarifications in the abstract and conclusions in response to the comments by Referee #1.

>Specific comments >1. Line 16 of page 2, the paper of Shen et al.[2008,AG] and

Forsyth et al.,(2009,AG) should be cited when referring the disturbance of solar wind flow as the flapping source. Shen, C., Z. J. Rong, X. Li, M. Dunlop, Z. X. Liu, H. V. Malova, E. Lucek, and C. Carr (2008), Magnetic configurations of tail tilted current sheet, Ann. Geophys., 26, 3525–3543. Forsyth, C., M. Lester, R. C. Fear, E. Lucek, I. Dandouras, A. N. Fazakerley, H. Singer, and T. K. Yeoman (2009), Solar wind and substorm excitation of the wavy current sheet, Ann. Geophys., 27, 2457–2474.

We can add these references.

>2. The joint observation of flapping event by TC-1 and Cluster (Zhang et al., 2005,AG) showed that the kink-like flapping waves propagate longitudinally with the same flapping phase at different X coordinates. However, it can not be characterized in your 2-D simulation, e.g. Fig1. Zhang, T. L., et al. (2005), Double Star/Cluster observation of neutral sheet oscillations on 5 August 2004, Ann. Geophys., 23, 2909–2914.

Considering the 1 min temporal resolution and relatively small (5 RE) separation in x direction between the satellites in the study by Zhang et al. (2005), we do not see that there would be a discrepancy between our results and theirs.

>3. Fig.3 predicts a dispersive flapping waves with time-increased frequency. To my knowledge, there is no observation evidence to back up it. Careful comparisons are needed.

We agree with the Referee and indeed suggest further studies to validate the simulation results against observations (page 7, lines 25-26).

>4. Even your 2-D simulation is valid to explain the triggering of kink-like flapping, the magnetopause disturbance is not the unique source. What I mean is that, the sources result in the pressure imbalance over tail current sheet could be multiple.

We agree with the Referee. It is quite likely that occasionally there would be multiple hemispherically asymmetric signatures that could all result in a displacement of the current sheet. The result would probably be more irregular flapping signatures, which

is not in disagreement with observations. We suggest to add on page 7, line 23: "Any hemispherically asymmetric magnetopause perturbation could cause tail flapping as shown here, but the shown perturbation initiated by subsolar magnetopause reconnection is a good example of a simulationally confirmed perturbation which indeed does cause this."

---

## Referee Comment (RC5) · Anonymous Referee #2 · 27 Jun 2018

1. As you replied that your 2-D simulation cannot definitely differentiate which flapping type it is, it is only a potential candidate to explain the source to trigger the kink-like flapping, thus I think the title of your paper could be better changed as "A possible source mechanism for magnetotail flapping motion...". 2. Although you calculated the dBx/dt, Vz, the location of plasma sheet, and shown it in Figure 6, I did not see any comparisons between your simulation and the actual observation properties of flapping motion. I understand your simulation is 2-D, you are unable to compare the wavelength, propagation speed, etc., but you can compare the flapping period at least. From your Figure 3, Figure 6, your flapping period is about 2 hours, it is evidently much larger than the typical observed flapping period (10 mins). The simulation is a good tool to explore the physical mechanism, but I CANNOT accept it without any comparison with

observations. 3. As you agree with my comment that, the source could be multiple. Here, you only consider the case of solar wind that "Steady solar wind, characterized by Maxwellian distribution functions, proton density of 1 cm$-$3, temperature of 0.5 MK, velocity of $-750$ km/s along the x axis, and magnetic fideld of $-5$ nT along the z axis (purely southward IMF)". Have you considered the other solar wind conditions, e.g. the northward IMF; the SW with a jump of dynamic pressure? I think you have to answer a question if your study is really important: Among the possible multiple sources, how much the case you studied contribute to the tail flapping motion?

---

## Author Comment (AC3) · 27 Jun 2018

>1. As you replied that your 2-D simulation cannot definitely differentiate which flapping type it is, it is only a potential candidate to explain the source to trigger the kink-like flapping, thus I think the title of your paper could be better changed as "A possible source mechanism for magnetotail flapping motion...".

Please note that the title does not only refer to the kink-like flapping in the y direction but also flapping in the x-z plane. While the significance of our results as a source mechanism to the waves in y direction remains a point of discussion, we have shown that in the simulation the mechanism does indeed produce flapping in the x-z plane.

Nonetheless, we can change the title to "A possible source mechanism for magnetotail

current sheet flapping".

>2. Although you calculated the dBx/dt, Vz, the location of plasma sheet, and shown it in Figure 6, I did not see any comparisons between your simulation and the actual observation properties of flapping motion. I understand your simulation is 2-D, you are unable to compare the wavelength, propagation speed, etc., but you can compare the flapping period at least. From your Figure 3, Figure 6, your flapping period is about 2 hours, it is evidently much larger than the typical observed flapping period (10 mins). The simulation is a good tool to explore the physical mechanism, but I CANNOT accept it without any comparison with observations.

Please note that the time in the figures in not given in HH:MM (hours and minutes) but MM:SS (minutes and seconds). This information is provided both in the caption of Figure 1 and on page 3, line 24 where this figure is first introduced. The captions of the following figures provide the information that they follow a format similar to that of Figure 1. However, as the notation was not clear, we are happy to include the explanation ", where MM indicates minutes and SS seconds" both on page 3, line 24 and in the caption of Figure 1.

We review many observational properties of current sheet flapping, including numerical values, in the Introduction. In section 3.1 we compare these numbers (including Bx and Vz) with those from our simulation, and find good agreement. The flapping period is compared with the observations of Sergeev et al. (1998) on page 3, lines 30-31.

>3. As you agree with my comment that, the source could be multiple. Here, you only consider the case of solar wind that "Steady solar wind, characterized by Maxwellian distribution functions, proton density of 1 cm$-3$, temperature of 0.5 MK, velocity of $-750$ km/s along the x axis, and magnetic field of $-5$ nT along the z axis (purely southward IMF)". Have you considered the other solar wind conditions, e.g. the northward IMF; the SW with a jump of dynamic pressure? I think you have to answer a question if your study is really important: Among the possible multiple sources, how

much the case you studied contribute to the tail flapping motion?

A particular set of solar wind conditions instead of a range of values was used because of the heavy computational load of running this type of a simulation. This same run has been analyzed in several previous studies as well (page 2, lines 23-25). The run was suitable for our study, because current sheet flapping occurred in it.

The question posed by the Referee is certainly interesting and a good topic for a future study. However, we consider it to be outside of the scope of the present paper.

---

## Referee Comment (RC6) · Anonymous Referee #2 · 28 Jun 2018

I have no further comments.

I'm eager to see the revised paper as soon as possible.

To make the introduction full, two papers about the empirical flapping models could be cited for the better, and it could be beneficial for your future 3-D simulation. 1.Petrukovich, A.A., Baumjohann, W., Nakamura, R., Runov, A., 2008. Formation of current density profile in tilted current sheets. Ann. Geophys. 26, 3669–3676. 2.Rong, Z. J., C. Shen, A. A. Petrukovich, W. X. Wan, and Z. X. Liu (2010), The analytic properties of the fiCapping current sheets in the Earth magnetotail, Planet. Space Sci., 58(10), 1215–1229, doi:10.1016/j.pss.2010.04.016.

[Figure]

2018.

---

## Author Comment (AC4) · 11 Jul 2018

We thank the referee for the comments and will proceed with the changes.

On behalf of the authors, Yann Pfau-Kempf

———————————————————

---

## Author Comment (AC5) · 11 Jul 2018

We thank the referee for the comments and will proceed with the changes.

On behalf of the authors, Yann Pfau-Kempf

---

## Author Comment (AC6) · 11 Jul 2018

We thank the referee for the comments and will proceed with the changes.

On behalf of the authors,

Yann Pfau-Kempf